# Modelling Hyperglycaemia in an Epithelial Membrane Model: Biophysical Characterisation

**DOI:** 10.3390/biom12101534

**Published:** 2022-10-21

**Authors:** Ana Reis, Joana P. F. Teixeira, Ana M. G. Silva, Mariana Ferreira, Paula Gameiro, Victor de Freitas

**Affiliations:** REQUIMTE/LAQV, Departamento de Química e Bioquímica, Faculdade de Ciências, Universidade do Porto, Rua do Campo Alegre, 4169-007 Porto, Portugal

**Keywords:** lipid glycation, liposomes, anisotropy, dynamic light scattering, Amadori products

## Abstract

Biomimetic models are valuable platforms to improve our knowledge on the molecular mechanisms governing membrane-driven processes in (patho)physiological conditions, including membrane permeability, transport, and fusion. However, current membrane models are over simplistic and do not include the membrane’s lipid remodelling in response to extracellular stimuli. Our study describes the synthesis of glycated dimyristoyl-phosphatidylethanolamine (DMPE-glyc), which was structurally characterised by mass spectrometry (ESI-MS) and quantified by NMR spectroscopy to be further incorporated in a complex phospholipid (PL) membrane model enriched in cholesterol (Chol) and (glyco)sphingolipids (GSL) designed to mimic epithelial membranes (PL/Chol/GSL) under hyperglycaemia conditions. Characterisation of synthesised DMPE-glyc adducts by tandem mass spectrometry (ESI-MS/MS) show that synthetic DMPE-glyc adducts correspond to Amadori products and quantification by ^1^H NMR spectroscopy show that the yield of glycation reaction was 8%. The biophysical characterisation of the epithelial membrane model shows that excess glucose alters the thermotropic behaviour and fluidity of epithelial membrane models likely to impact permeability of solutes. The epithelial membrane models developed to mimic normo- and hyperglycaemic scenarios are the basis to investigate (poly)phenol-lipid and drug–membrane interactions crucial in nutrition, pharmaceutics, structural biochemistry, and medicinal chemistry.

## 1. Introduction

Changes to people’s lifestyles and eating habits in the last decades have had a detrimental impact on human health. In the past five decades, the incidence of diabetes and obesity in the worldwide population has tripled [1], but even more alarming is the rising prevalence of diabetes among young adults [2,3]. In view of the endothelial dysfunction associated with diabetes and complications to highly vascularised organs (heart, kidney, and brain), the socio-economic burden of diabetes is expected to escalate in an increasingly ageing population, contributing to rising morbidity and mortality worldwide.

At the cellular level, endothelial dysfunction is accompanied by the release of pro-inflammatory cytokines and chemokines with disruption of endothelial cell bioenergetics [4,5,6]. In addition to this, in vitro cellular studies in human endothelial umbilical vein endothelial cells have equally shown that hyperglycaemia alters the biophysical properties of epithelial membranes by stiffening of the lipid bilayer [7]. Others working with model membranes found that the enrichment of phospholipid (PC/PE) liposomes with glycated phosphatidylethanolamine (Glyc-PE), a by-product found in circulation in diabetic patients [8,9], led to a significant decrease of membrane potential which, according to the authors, could have contributed to the increased uptake of liposomes by Caco-2 model cells [10].

While in recent years there is an increasing effort to integrate biological findings with biophysical data to improve our understanding on the molecular and cellular mechanisms underlying the onset, progression, and treatment of endothelial dysfunction in diet-related diseases, to date findings from biomimetic models have had little translation to (patho)physiological processes. While biomimetic models can be tailored to better mimic the lipid composition of cell membranes, in reality, current biomimetic models are over simplistic as they (i) do not reflect the diversity of lipid classes of epithelial cell membranes [11,12], (ii) do not account for the cholesterol-to-phospholipid (Chol/PL) ratios of cells membranes and organelles [13], and finally (iii) do not include the lipid remodelling that takes place with age and (patho)physiological conditions, namely the presence of oxidative lipid modification typical of chronic inflammatory states and that impact membrane biophysical properties [10,14,15]. Interestingly, some works have tentatively included higher lipid complexity [16,17,18,19,20] by the inclusion of cholesterol and glycosphingolipids typical of epithelial membranes [13], though none of those investigated the impact of lipid glycation on the biophysical properties of membrane models.

Following our group’s interests in investigating the role of dietary (poly)phenols on the membrane organisation, fluidity, and impact on the permeability of solutes (metabolites, gases, drugs) across the endothelial bilayer in (patho)physiological conditions, we built a biomimetic model with complex lipid composition (PL/Chol/GSL) bearing lipid environments (Chol/PL ratio) characteristic of epithelial membranes [13] and tailored to hyperglycaemia conditions by the incorporation of synthetic glycated dimyristoyl phosphatidylethanolamine (DMPE-glyc). The biophysical properties of epithelial membrane models under normo- and hyperglycaemic conditions were evaluated.

## 2. Materials and Methods

### 2.1. Reagents

Lipids 1,2-stearoyl-sn-glycero-3-phosphocholine (DSPC), 1,2-palmitoyl-sn-glycero-3-phosphocholine (DPPC), 1,2-myristoyl-sn-glycero-3-phosphoethanolamine (DMPE), and glucosyl-β1-1′-N-stearoyl-D-erythro-sphingosine (C18-GlcCer, GSL) were purchased from AVANTI Polar LIPIDS (Birmingham, AL, USA). Cholesterol (Chol) was purchased from Sigma Aldrich (Madrid, Spain). Structure of lipids are shown in Appendix A. Fluorescent probe 1,6-diphenyl-hexatriene (DPH) was purchased from Sigma, phosphate buffer saline (PBS) tablets were purchased from PanReac AppliChem (Darmstadt, Germany).

### 2.2. Synthesis of DMPE-Glucose Adducts

Modification of DMPE by glucose was carried out in 10% (*v*/*v*) aqueous methanol according to previously described conditions [21,22]. Briefly, 10 mg of DMPE (15.74 μmoles) were solubilised in methanol with a few drops of ethanol and placed into a sovirel glass tube. Glucose (141 mg), corresponding to a 50 molar excess of glucose (787 μmoles), was dissolved in water and added to the DMPE solution (final concentration 2 mg/mL). The mixture was homogenized in a vortex (1 min) and left to react at 80 °C in a heating block (Stuart, SBH 130D/3, Stone, Staffordshire, UK) for 5 h. After this time, excess of unreacted glucose was washed by liquid–liquid washing steps using mixture of chloroform:water (2:1, *v*/*v*) to achieve phase separation. Organic extract containing DMPE-glyc mixed with DMPE was dried under vacuum (LabConco, Kansas City, MI, USA) at room temperature until dryness. Replicates (triplicates) of dried extracts were structurally characterised and kept in inert nitrogen atmosphere at −20 °C until further incorporation in LUVs.

### 2.3. Characterisation of DMPE-Glucose Adducts by Mass Spectrometry (ESI-MS) and Nuclear Magnetic Resonance (NMR) Spectroscopy

#### 2.3.1. ESI-MS and ESI-MS/MS Analysis

The dried organic extract was qualitatively and quantitatively characterised. The presence of DMPE-glyc adduct in dried extracts was confirmed by electrospray mass spectrometry (ESI-MS) and quantification of the DMPE-glyc adduct in the mixture was achieved by proton nuclear magnetic resonance (^1^H NMR) spectroscopy. Mass spectrometry characterisation was conducted by direct injection in a LCQ DECA XP ion trap mass spectrometer (Thermo Fischer Scientific, Bremen, Germany) equipped with an atmospheric pressure ionisation (API) source using electrospray ionisation (ESI) interface (ESI-(IT)-MS). Dried extract was solubilized in methanol (0.1% formic acid, *v*/*v*) and analyzed in the positive ion mode. The capillary voltage was maintained at 5 kV and cone voltage was set at 15 V; source temperature set at 270 °C and sheath gas flow at 19 (arbitrary units). Samples were introduced into the electrospray source at a flow rate of 18 μL/min. Each spectrum was produced by accumulating data for 1–2 min. Tandem mass spectra (ESI-MS/MS) was obtained using nitrogen as the collision gas and the collision energy used for protonated molecule ([M+H]^+^) was set at 25 V (arbitrary units). The raw data were processed and transformed into values of molecular masses using Excalibur software (San Jose, CA, USA, version 2.2).

#### 2.3.2. NMR Analysis

Quantification of DMPE-glyc adduct in the reaction mixture was carried out by NMR experiments. For the proton NMR analysis, the dried extract was dissolved in 0.6 mL of CD_3_OD (d_4_, 0.03% TMS; Eurisotop, Saint Aubin, France) and transferred into a 5 mL NMR tubes (NOR509UP7, Norell Series, Sigma, St. Louis, MI, USA) and NMR measurements started within 1 h. The NMR spectra of DMPE were recorded on 400 MHz NMR spectrometer (operating at 400.15 MHz for protons) and for DMPE-glyc reaction mixture on 600 MHz NMR spectrometer (operating at 600.13 MHz for protons). CD_3_OD was used as the solvent and tetramethylsilane (TMS) as the internal reference; the chemical shifts are expressed in δ (ppm) and the coupling constants (J) are in Hz. The two-dimensional ^1^H/^1^H correlation spectra (COSY) were acquired using the standard Bruker software. Bruker’s integrated software (Topspin version 3.2.3 for 400 MHz and version 3.6.2 for 600 MHz spectrometers; Ettlingen, Germany) includes data acquisition and processing. The COSY analysis was carried out with a data size of 2K (F2) × 256 (F1) points for a spectral width of 13.350 ppm, using FID resolution of 41.733414 Hz for the 400 MHz and 62.600159 Hz for the 600 MHz spectrometers.

### 2.4. Preparation of Large Unilamelar Vesicles (LUVs)

Large unilamellar vesicles (LUVs) were freshly prepared for spectroscopic measurements and protected from light prior to use. Briefly, lipids were dissolved in chloroform-methanol (1:1, *v*/*v*) mixed in a round-bottom glass flask. For anisotropy experiments, fluorescent probe 1,6-diphenyl-1,3,5-hexatriene (DPH) dissolved in chloroform (stock solution 1.69 mM) was added to the lipid organic mixture in a lipid:probe ratio of 300:1 (mol/mol). Lipids were evaporated to dryness under vacuum for 3 h to remove all traces of organic solvent. Liposomes were formed by the hydrating–extrusion method [23] by the addition of phosphate buffer (10 mM, pH 7.4) to a final lipid concentration of approximately 1 mM (0.65–0.70 mg/mL). Multilamellar vesicles (MLVs) were processed by interchangeable freeze/thaw cycles (10) in liquid nitrogen and boiling water bath. Lipid suspensions were vortexed and extruded in a LIPEX Biomembranes extruder attached to a CLIFTON thermostatic circulating water bath above the phase transition temperature (70 °C) using polycarbonate filters (100 nm pores, Whatman^®^, Maidstone, UK) to obtain large unilamellar vesicles (LUVs). The mean diameter of LUVs was monitored by dynamic light scattering (Zetasizer Nano ZS, Malvern Instruments, Malvern, UK). The LUVs were kept at 4 °C until spectroscopic characterisation.

### 2.5. Characterisation of Thermotropic Behaviour of Epithelial Models by Spectroscopic Approaches

The thermotropic behaviour of model LUVs with increasing complexity was carried out by two complementary spectroscopic approaches namely by steady-state fluorescence (anisotropy) using fluorescent labelled liposomes with DPH probe and by dynamic light-scattering spectroscopy using unlabelled liposomes. Anisotropy measurements were performed in a Varian Cary Eclipse spectrofluorometer, equipped with a temperature cell holder (Peltier Multicell Holder, Cary Temperature Probe series II, Varian, VIC, Australia) coupled to a water-circulation thermostat. For all systems studied, liposomes were placed in 1.5 mL quartz cuvettes and steady-state fluorescent anisotropy measurements were recorded in the temperature range from 20–80 °C with 2 °C increments and maintained for 3 min before each measurement. Temperature of samples was monitored using an in-house built sensor immersed inside the quartz cuvette. The excitation (λexc) and emission wavelengths (λem) used were 360/427 nm and slit widths set to 5 nm for both excitation and emission values. Anisotropy values (r_s_) are defined by Equation (1):(1)$$r_{s}=\frac{I_{VV}-I_{VH}G}{I_{VV}+2\;I_{VH}G}$$
where I*_VV_* and I*_VH_* represent the emission intensities when the polarisers were oriented vertical–vertical (parallel) and vertical–horizontal (perpendicular) to the excitation beam. *G* is a correction factor and is given by the ratio of vertical to horizontal components when the excitation light is polarised in the horizontal direction, G = I_HV_/I_HH_ [24]. Anisotropy values shown are the mean of three independent measurements (n = 3). Raw r_s_ data was analyzed in Origin (version 9.0), as a function of temperature and the following equation (Equation (1)) was fitted to the experimental data using Equation (2):(2)$$r_{s}=r_{s2}+\frac{r_{s1}-r_{s2}}{1+10^{B^{\prime }\left(\frac{T}{T_{m-1}}\right)}}$$
where T is the absolute temperature, T_m_ is the midpoint phase transition, and r_s1_ and r_s2_ are the upper and lower values of r_s_; B’ is the slope factor, correlated with the extent of cooperativity (B), given by B = [1 − 1/(1 + B’)]; the introduction of B yields a convenient scale of cooperativity ranging from 0 to 1 [25]. Information on the thermotropic behaviour of LUVs was further complemented by dynamic light scattering measurements acquired in a Zeta Sizer Nano ZS instrument (Malvern Instruments, Malvern, UK). The measurements of mean count rate versus temperature were carried out in capped 3.5 mL glass cuvettes (Starna Scientific, Hainault Essex, UK) using a helium–neon laser (λ = 633 nm) at an angle of 173° with fixed attenuation (6) over a temperature range from 20 to 80 °C with 2 °C increments. Samples were left to equilibrate for 3 min at each set value before measurements. Raw mean count rate data were analyzed in Origin (version 9.0) as a function of temperature. The first-derivative plot of the experimental data was performed to determine the transition temperatures of each system.

### 2.6. Determination of Surface Potential

The net surface charge of epithelial membrane model under normo- and hyperglycaemia conditions was calculated using electrophoretic mobility measured in a Malvern Zetasizer PanAnalytical Nano ZS device (Malvern Instruments, Malvern, UK) at 37 °C with light detection at 17° using disposable plastic cells purchased from Malvern (PanAnalytical, Malvern, UK). Each sample was measured in triplicate, where each measurement was the average of 20 readings. Zeta potentials was calculated from electrophoretic mobility (UE) raw data is converted into zeta potential (ζ) using the Henry equation (Equation (3)) with the Smoluchowski approximation (software version 7.13, Malvern, UK), with ε as the dielectric constant of the dispersant, F(κa) is the Henry function, and η is the viscosity.
(3)$$UE=\frac{2\varepsilon \zeta F{\left(\kappa a\right)}_{}}{3\eta }$$

## 3. Results

### 3.1. Synthesis and Structural Characterisation of Glycated Dimyristoyl-Phosphatidylethanolamine (DMPE-Glyc) Adducts

Glycation of aminophospholipid dimyristoyl-phosphatidylethanolamine (DMPE) was carried out to mimic hyperglycaemia-induced modifications to epithelial membrane lipids. The in vitro non-enzymatic glycation reaction yield depends on several parameters, such as solvent composition, temperature, and reaction time [21,22,26], where lower temperatures usually require longer reaction times. Based on the previous published literature, the reaction conditions in this work were performed for 5 h at 80 °C trying to maintain a good reaction yield. The synthesised DMPE-glyc adduct containing the unmodified PE was not further purified to avoid extensive sample handling thus minimising the formation of degradation products due to increased susceptibility of glycated PE to oxidative modification when compared to unmodified PE [21,27]. The presence of DMPE-glyc adduct in the reaction mixture was qualitatively and quantitatively confirmed by mass spectrometry (ESI-MS) and nuclear magnetic resonance (NMR) spectroscopy, respectively.

The ESI full mass spectrum of the reaction mixture (Figure 1A) shows the presence of ions at *m*/*z* 636.5 attributed to the protonated molecule of the unmodified DMPE ([DMPE+H]^+^), the ion at *m*/*z* 658.4 attributed to the sodium adduct of the unmodified DMPE ([DMPE+Na]^+^) and ion at *m*/*z* 798.2, consistent with the protonated molecule [M+H]^+^ of DMPE-glyc adduct. The product ion mass spectrum of protonated DMPE-glyc adduct ([M+H]^+^, *m*/*z* 798.4) (Figure 1B) exhibits the product ion at *m*/*z* 495.2 formed by loss of 303 a.m.u. from the precursor ion attributed to the combined loss of glucose (162 a.m.u.) and phosphatidylethanolamine moiety (141 a.m.u.). The loss of neutral molecule with 303 a.m.u. is characteristic of protonated glycated PE molecules [9,21,28], confirming the covalent linkage of glucose to the amino group of PE. The absence of fragment ion formed by loss of 120 a.m.u. (*m*/*z* 678) from the precursor ion and characteristic of Schiff-base adducts [27] confirms the presence of DMPE-glyc in the reaction mixture as Amadori adduct (Figure 1).

The quantification of DMPE-glyc in the reaction mixture was conducted by NMR experiments, including ^1^H NMR and two-dimensional ^1^H/^1^H correlation spectrum (COSY) (Figure 1C,D). For comparison purposes the ^1^H NMR and two-dimensional ^1^H/^1^H correlation (COSY) spectra were also carried out for the unmodified DMPE (Appendix A). The ^1^H NMR spectrum of unmodified DMPE (Appendix A) reveals that the most deshielded signal is a multiplet at 5.21–5.26 ppm attributed to the H-2 of the glycerol backbone, suffering a strong deshielding effect due to the two ether groups and the phosphate group present in the molecule. Based on the COSY spectrum (Appendix A), this multiplet is correlated with the following signals: (i) a triplet at 4.00 ppm due to two methylene protons (H-3) and (ii) two doublets of doublets at 4.18 and 4.44 ppm attributed to the other methylene protons (H-1). A multiplet signal appears at 4.02–4.06 ppm, which corresponds to H-4 of the ethanolamine group. Hence, H-4 is correlated with a triplet at 3.16 ppm, which is assigned as H-5. A doublet of doublets at 2.33 ppm is due to H-6 and H-7, which are correlated with the multiplet at 1.59–1.64 ppm, attributed to methylenic protons of the fatty acid chain. A multiplet at 1.28–1.31 ppm and a triplet at 0.90 ppm complete the last CH_2_ and CH_3_ protons of the fatty acid chain. This analysis is in good agreement with previous data reported by Wang and Hollingsworth [29]. Although the proton signals obtained for the DMPE-glyc extract (Appendix A) are similar to those of the unmodified DMPE (Appendix A), there are three important exceptions: (i) a group of signals appear from 3.60 to 4.00 ppm (Figure 1C), matching perfectly with those of the pyranosyl structure of the sugar residue from the Amadori adduct [26]; (ii) a multiplet at 4.10–4.14 ppm, which was attributed to H-4′ protons of the DMPE-glyc adduct; and (iii) this multiplet shows correlation with a multiplet at 3.26–3.28 ppm due to the H-5′ of the adduct (Figure 1D). From the integration of the more isolated signal at 3.16 ppm corresponding to two H-5 protons of the starting DMPE molecule and the 4.10–4.14 ppm signal corresponding to two H-4′ protons of the DMPE-glyc adduct, it is possible to quantify the DMPE-glyc adduct in the mixture. Thus, considering the ratio of H-5_DMPE/H-4′_adduct obtained in the three replicates performed (Appendix A and Figure 1), the average percentage of synthesised DMPE-glyc adduct calculated in the mixture is 8.7 ± 0.2% of total PE (*n* = 3) showing good reproducibility for the lipid glycation reaction.

In this study, although the transition temperature of synthesised DMPE-glyc was not determined; it is expected to be slightly lower than T_m_ = 55 °C (transition temperature of unmodified DMPE), as previous DSC studies showed the transition temperature of glycated DPPE adduct at T_m_ = 62.6 °C, which was slightly lower than that of the corresponding unmodified DPPE at T_m_ = 63.3 °C [27].

### 3.2. Development and Thermotropic Characterisation of Epithelial Membrane Model (PL/Chol/GSL) under Normoglycaemic Conditions

In this study, liposomes of complex composition (PL/Chol/GSL) containing phospholipids (PL) together with cholesterol (Chol) and glucosphingolipids (GSL), were built to reflect the unique lipid composition of epithelial plasma membranes (mouth, stomach, kidney, intestine, lung, endothelial, others) with high cholesterol-to-phospholipid ratio [13]. Our model incorporates choline (PC) and ethanolamine (PE) lipids as these classes predominate in mammalian epithelial cell membranes [30,31,32]. Glycosphingolipids (GSL), specifically glucosyl-ceramides, were selected as these predominate in epithelial cell lines over galactosyl-ceramides [33]. While the fatty acid composition of plasma epithelial membranes contains saturated and unsaturated acyl chains [11,32], the choline and ethanolamine phospholipids selected for this study contain saturated acyl chains to minimise oxidative degradation via radical-mediated reactions [34] during handling and LUV preparation.

The thermotropic behaviour of epithelial modelled liposomes (PL/Chol/GSL, 0.59:0.23:0.17; mol%) was investigated by steady-state fluorescence anisotropy using liposomes labelled with DPH fluorescent probe. As steady-state anisotropy fluorescence experiments involve the addition of an external fluorescent probe and provide an indirect measurement of the changes “sensed” by the fluorescent probe, dynamic light scattering (DLS) experiments using unlabeled liposomes were also carried out to complement the DPH anisotropy measurements. As shown from the anisotropy and the mean count rate plots obtained for the phospholipid system (Figure 2), both plots show a remarkable similar behaviour for the binary PC system shown in Figure 2A,B and the ternary PC/PE model shown in Figure 2C,D exhibiting a clear fluid-to-gel transition. Incorporation of the aminophospholipid (DMPE) into PC system (DPPC/DSPC, 0.65:0.35, mol%) has no visible effect on the gel-to-fluid transition displaying remarkable similarity between fluorescence anisotropy and DLS datasets thus corroborating the applicability of these approaches in complex ternary systems.

Although anisotropy readings gives a measurement of the rotational motion of DPH probe localised deeply within the membrane bilayer [35,36] and DLS measures changes in optical properties of systems occurring during phase transitions detected by differences in the scattering intensity of samples through measurements of the count rates [37], both spectroscopic approaches have been previously applied to the characterisation of transition temperatures of single [37] and multicomponent liposomes [38,39] highlighting its complementary character. In fact, values of mean transition temperatures obtained for the PL system with complex composition (Table 1) estimated from the raw mean count rate data from DLS (first-order derivatives) for the systems studied are within the same order of magnitude as those determined by steady-state fluorescence (Table 1) supporting the complementary character of both spectroscopic techniques in the characterisation of complex models offering an alternative to equally simple techniques such as differential scanning calorimetry [27,40,41] and other advanced approaches such as atomic force microscopy [42], ESR [43], and nanoplasmonic sensors [44]. Moreover, as preparation of liposomes for DLS experiments do not require the incorporation of exogenous fluorescent probe, this non-invasive technique offers an added advantage over the anisotropy fluorescence approach.

The DPH fluorescence anisotropy plots obtained for the liposome models with increasing complexity (shown in Figure 3) show that the inclusion of cholesterol (Chol) into the phospholipid systems (PL/Chol and PL/Chol/GSL) broadens the gel–fluid transition. The DPH anisotropy plot of PL/Chol system shows that while there is a continuous decrease in anisotropy as temperature increases, DPH anisotropy values are higher when compared to PL system above the transition temperature. The increase of DPH anisotropy with temperature is indicative of membrane stiffening (“rigidification”) in Chol-rich liposomes [43,45,46] masking the gel-to-fluid transition (Table 1).The presence of high-Chol content in endothelial cells is crucial for their barrier properties and may serve as a mechanism by which cells cope with stress as Yamamoto and colleagues reported the incorporation of Chol in plasma membranes of stress stimulated endothelial cells with consequent membrane stiffening [47]. The incorporation of Chol, together with the incorporation of PE lipids, is thought to be one of the mechanisms by which cells regulate membrane fluidity and proper protein function for membrane-mediated processes [48]. Similarly, incorporation of GSL in the PC/Chol model (PL/Chol/GSL) shown in Figure 3 led to a slight increase of anisotropy across all the temperature range when compared to PL/Chol model, in agreement with previously published findings [18,19] and with changes in transition temperatures (Table 1).

The three transition temperatures observed for PL/Chol system are likely to be attributed to PC/PE, PC/Chol, and PC/PE/Chol regions as cholesterol has low affinity with PE lipids [49] reflecting the coexistence of gel phase (S_o_) with Chol-poor (liquid disordered-S_o_) domains along with Chol-rich PL (liquid ordered-I_o_) domains [42]. Previous fluorescence microscopy experiments conducted on PL/Chol mixtures at low mol% Chol (below 30 mol%) with high melting lipids (such as those found for saturated PC and PE) showed the coexistence of liquid ordered (l_o_), liquid-disordered (l_d_) and I_α_ domains visible at the surface of GUV over a range of lipid composition and temperatures [50].

For the PL/Chol/GSL system, the occurrence of three transition temperatures suggests the heterogeneous character of membrane domains composed of gel-like PC/PE domains, as well as PL/Chol liquid ordered domains in PL liquid disordered phase and GSL/Chol regions [18,19]. Although PC, Chol and GSL are highly immiscible under near physiological conditions (high PL content and low-to-medium Chol and GSL content [18]) C18-GlcCer can, nevertheless, interact with Chol-forming lo phase [51] forming gel-phase domains in the fluid PC regions [52]. Considering that membranes with high melting lipids (saturated PC and PE) and cholesterol contain more lo domains [50] fluorescence microscopy experiments conducted in GUVs with the predominance of liquid ordered domains (l_o_) bearing 20 mol% of C16:0-GlcCer at neutral pH shows round shaped domains from where the fluorescent probes were excluded suggesting the interaction of C16:0-GlcCer with Chol in more tightly packed lo phases [18].

### 3.3. Effect of Hyperglycaemia on the Properties of Epithelial Membrane Model (PL/Chol/GSL)

Under in vivo hyperglycaemia conditions, the excess of glucose in circulation is responsible for the modification of free amino groups present in glycerophosphatidylethanolamines (PE) by non-enzymatic cross-linking reaction leading to the formation of glycated lipids (Figure 1) that affect the properties of cell membranes and in turn impact membrane-related processes [7,10,27].

Hyperglycaemic conditions were mimicked in the epithelial membrane model by the incorporation of synthesised DMPE-glyc adduct in the epithelial membrane model (PL/Chol/GSL). The amount of DMPE-glyc incorporated in the epithelial membrane model was circa 4% of total PE bearing in mind previous values reported in human plasma [8,9]. The epithelial liposomes tailored to hyperglycaemic conditions (DPPC/DSPC/DMPE/DMPE-glyc/Chol/GSL) were characterised according to their thermotropic behaviour, size, surface potential and anisotropy, and results are summarised in Table 2. For comparison purposes, Table 2 also summarises the properties of epithelial LUVs mimicking normoglycaemic conditions (DPPC/DSPC/DMPE/Chol/GSL).

The thermotropic behaviour of hyperglycaemic epithelial model was studied by DLS in view of the previous similarity between steady-state fluorescence values and DLS values under normoglycaemia (Table 1). The mean count rate plots show that incorporation of DMPE-glyc impacted the thermotropic behaviour of the epithelial membrane model (PL/Chol/GSL) under hyperglycaemic conditions (Figure 4B,C).

The first derivative graph (Figure 4C) shows a subtle transition at low temperatures (t = 39.3 °C) and a second clear transition at t = 53.8 °C (Table 2) which could be related to the presence of DMPE-glyc. Previous DSC heating experiments revealed that glycation of PE resulted in the reduction of the energy necessary to induce transition from a gel to a liquid disordered phase and in consequence a decrease in the transition temperature of modified saturated DPPE lipids [27]. Interestingly, other studies conducted on biomimetic models containing naturally glycosylated sphingolipids bearing the same ceramide moiety and differing only in the presence of glucose residue (C16-Cer and C16-GlcCer), showed remarkable differences in their partial phase diagrams [18,51]. The presence of glucose residue was thought to increase the interactions with neighboring lipids and responsible for the increase miscibility of the ceramide moiety in the lo phase. In our study, the cross-linking modification of DMPE with glucose sugar residues may facilitate the interaction of glycated DMPE with glucosylsphingolipids (GSL) and the formation of gel domains in fluid PL phase of more complex systems in spite of GSL having a high melting temperature [53] and a strong tendency to segregate into tightly packed gel domains [18,52]. In fact, previous studies mentioned that PE is likely to form ordered domains with sphingosine-based lipids [54,55].

Interestingly, the incorporation of low concentrations DMPE-glyc (circa 4 mol%) resulted in a slight decrease of DPH anisotropy at physiological temperature (Table 2). The decrease in DPH anisotropy may be attributed to the incorporation of DMPE-glyc in the lipid bilayer and not to other lipid degradation product such as the formation of oxidised phosphatidylcholines (oxPC) which are described to disrupt lipid packing, increase membrane deformability, water penetration and barrier disruption in membrane models [56,57,58,59] and in endothelial cells [14]. The decrease in DPH anisotropy at physiological temperature observed by the incorporation of small amounts of DMPE-glyc (~4 mol%) in the lipid bilayer (Table 2) results in increased membrane fluidity under hyperglycaemia conditions. The increase in membrane fluidity observed in our study under hyperglycaemia is in slight contrast with others describing that diabetes induced an increase in membrane stiffness in human umbilical vein endothelial cells [7]. Nevertheless, our results should be interpreted taking other factors into consideration, namely the decrease of PE/PC ratio in hyperglycaemia (Table 2) known to decrease membrane rigidity [48], secondly the absence of unsaturated PC and PE characterised by lower transition temperatures which would increase membrane fluidity, and finally the increased susceptibility of glycated PE to oxidative modification which can lead to carboxymethyl-PE (CM-PE), carboxyethyl-PE (CE-PE) and other cleavage products involving the glycated polar head [21,27]. To date, the biophysical changes induced by AGE in epithelial artificial membranes are not yet known, prompting further studies.

The incorporation of DMPE-glyc (4 mol%) into the complex epithelial membrane model had no apparent effect on the size of the liposomes or the surface net charge (Table 2). Even though the ζ values of the epithelial model in normoglycaemia are in agreement with previous experiments conducted in complex lipid models (PC/SM/Chol/GSL) at neutral pH [19], it is likely that these apparent unchanged values in hyperglycaemia are due to the low incorporation of DMPE-glyc (circa 4%) as in other studies the incorporation of higher amounts of glycated PE (as high as 50%) resulted in a decrease of the surface net charge to more negative values [10,27].

## 4. Discussion

While the incorporation of glycated PE had previously been studied in artificial membranes [10,27], our study stands out from others for various reasons: (1) by the inclusion of Chol and GSL, whereas others have conducted their studies in simple PL (PC/PE) models; (2) by the inclusion changes to cholesterol-to-phospholipid ratios (lipid environments) and PC/PE ratios occurring in diabetes [60]; and (3) by the inclusion of a more physiologically relevant amount of glycated PE [8,9], setting the basis for a more suitable membrane model. In spite of the increasing effort to complement cell-based findings membrane biophysical experiments and improve our understanding on the molecular mechanisms by which diabetes impacts membrane-driven processes, in reality, the scarcity of suitable membrane models reflecting the unique lipid composition of epithelial membranes bearing high cholesterol (Chol) and glycosphingolipid (GSL) ratios [11,12,61] has hindered the translation of in vitro findings to (patho)physiological conditions.

One of the reasons behind the scarcity of models mimicking epithelial membranes in general and the endothelium in particular is due to the poor knowledge of the endothelial lipidome [13] despite the recent technological advances in mass spectrometry and lipid analysis pipeline that boosted the analysis of biofluids in clinical lipidomics [62]. To the best of our knowledge, only three studies focused on the (phospho)lipidome profile of endothelial cells [63,64,65], having identified hundreds of molecular species by untargeted MS-based approaches and their changes under oxidative [63,64], glucotoxic [64], and mechanical conditions [65]. However, because the authors used MS-based approaches the findings reported are qualitative and not quantitative. Moreover, none of the three studies conducted sub-cellular fractionation steps and hence the (phospho)lipidome reported relates to the whole cell lipidome and not to the plasma membrane lipidome in endothelial cells.

One other reason is related to the poor knowledge of the levels of lipid glycation in circulation in diabetes. Ravandi and colleagues reported that the extent of PE lipid glycation in plasma samples of diabetic patients was 10–16 mol% (of total PE) similar to the overall content of glycated haemoglobin (9–15 mol%) in circulation [8]. However, in another work the authors reported that levels of glycated PE in plasma in diabetic individuals ranged from 0.15–0.29 (mol% of total PE) [9]. The discrepancy of values reported in the literature is quite high, and may be related to the analytical approaches adopted in the two studies with the highest values quantified using normal phase HPLC-(ESI)-MS [8], whilst the lowest values were estimated by reverse phase LC using more specific targeted MRM approaches [9]. In this study, our epithelial model was incorporated with 4 mol% of glycated PE (total PE) an average value between those reported in the literature. Nevertheless, the discrepancy between the physiological levels of glycated PE reported in the literature [8,9] and the values of glycated PE incorporated in membrane models (25–50%) [10,27] warrants further studies to ascertain the true levels of glycated PE and its effect on the biophysical properties of membrane models.

## 5. Conclusions

In this study, we describe the effect of hyperglycaemia on the biophysical properties of a complex Chol- and glycosphingolipid-rich model mimicking the unique composition of epithelial membranes. Data obtained in this study highlights the impact of excess circulating glucose on membrane thermotropic behaviour, surface potential, size and fluidity. This tentative epithelial membrane model can be further applied in the study of polyphenol–lipid interactions improving our understanding on the molecular mechanisms behind the benefits of (poly)phenol-rich diets on the membrane packing and fluidity, permeability, and transport across endothelial membranes.

## Data Availability

Raw data will be supplied upon request.

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
