# Peer review of "Modelling Hyperglycaemia in an Epithelial Membrane Model: Biophysical Characterisation"

_biomolecules, 2022, doi:10.3390/biom12101534_

Round 1

Reviewer 1 Report

I have analyzed the manuscript of Reis and coauthors. I think that this is interesting study with mechanistical design, but it can has a value for computer and chemical modeling of membrane processes during hyperglycemia. I have some comments which related predominantly with physiological side of the study.

1. The title is too ambigous and incorrect. Authors did not studied and modeled diabetes. I think that in title "diabetes" must be improved on "hyperglycemia".

2. Authors discussed and studied glycation of proteins. However, some other reactive compounds can modified membrane lipids, for instance, malonyl dialdehyde. I think that authors should broad further studies in this area and add information about other reactive molecules in hyperglycemia in Introduction and Discussion.

3. Authors in title made mention about polyphenol-lipid interactions. However, this question was not analyzed in introduction and these interactions were not analyzed in this study. I think that it can be removed from the study or this data should be add in Introduction, Results, etc.

4. In Figures 3 and 4, Tables 1 and 2 authors made several measurements. Which statistical methods was used for the analysis of differences significance?

5. The resolution and quality of Figure 1 need improving.

6. Why authors selected miristic acid? I think that fatty acids with more long carbon chain have more critical significance (palmitic acid) in the dislipidemia and membrane structure contexts.

7. In the study authors analyzed just glycation on amino-group in phosphatidylethanolamine but not for phosphatidylserine?

Thus, I think that the study is interesting but need some improvements and consideration before publication.

Author Response

Response to the reviewer was uploaded as pdf file

Reviewer 2 Report

Comment

Date: 06-10-2022

Reis et al. addressed very interesting findings in their research article entitle as “Modelling diabetes in an epithelial membrane model for 2(poly)phenol-lipid interaction studies”. The work is interesting, well designed, and informative to reader working in the domains. However, I recommend few suggestions before publication.

Comment 1: The unit in the sentence “Modification of DMPE by glucose was carried out in 10% aqueous methanol (v/v)” must be presented as 10% v/v. I recommend to correct “15,8 µmoles”.

Comment 2: In mass spectroscopy, what was the basis for selection of 0.1% formic acid in methanol and working temperature of 270 °C. The software details must be added with city, state and country name “Excalibur software (version 1062.2”. Section 2.3 must be separated into two different sections. It is confusing to reader. NMR experimental details is shortly described. However, I suggest to describe in details. How did you purify your sample before NMR analysis? Please provide details of the software “Bruker software”

Comment 3: In section, please check stock solution centration 1.69 mM“ and the ratio 300:1. Which method was adopted to remove the trace content of organic solvent from thin film? How did authors ensure its absence? How did author stabilize vesicle without using any edge activator or surfactant?. After filtration, what percentage of vesicles were found within SUV type? Please provide details of trapped and free drug after filtration?  City, state, and Country name are missing in “Zetasizer Nano ZS, Malvern Instruments” .

Comment 4: In section, provide the equation “Henry equation”. Did authors dilute the sample before assessment of zeta potential value? In figure 1, scale and visibility are not clear to reader. I recommend to replace these image with high resolution. In figure 3, the scale is very light in color.   

Comment 5: The report size and zeta potential must be presented as original report in term of peak and intensity. This will give other information.   

Comment 6: In table 2, the values of zeta potentials are extremely low. How did you stabilize vesicles for long term stability? Vesicles are physically less stable for drug leakage at storage temperature. How did authors manage in his vesicular formulation?

Comment 7: Authors described as “This tentative epithelial membrane model can be further applied in the study of polyphenol-lipid improving our understanding on the molecular mechanisms behind the benefits 477of (poly)phenol rich diets on the membrane packing and fluidity, permeability, and transport across endothelial membranes”. In this, how did authors avoid other interacting factor at surface adsorption, absorption and membrane fluidity for polyphenol extract and subsequent membrane behavior to conclude this statement? Justify

Author Response

Response to reviewer was uploaded as pdf file

Round 2

Reviewer 1 Report

All my questions were addressed, I think that the manuscript can be accepted for publication.

Reviewer 2 Report

Authors revised the manuscript as per the comments.